# On application of asymmetric Kan-like exact equilibria to the Earth magnetotail modeling

Daniil B. Korovinskiy[1], Darya I. Kubyshkina[1], Vladimir S. Semenov[2], Marina V. Kubyshkina[2], Nikolai V. Erkaev[3,4], and Stefan A. Kiehas[1]

[1]Space Research Institute, Austrian Academy of Sciences, Graz, Austria
[2]Saint Petersburg State University, St. Petersburg, Russia
[3]Institute of Computational Modelling, FRC "Krasnoyarsk Science Center" SBRAS, Krasnoyarsk, Russia
[4]Siberian Federal University, Krasnoyarsk, Russia

*Correspondence to:* Daniil Korovinskiy (daniil.korovinskiy@oeaw.ac.at)

**Abstract.** A specific class of solutions of the Vlasov-Maxwell equations, developed by means of generalization of the well-known Harris-Fadeev-Kan-Manankova family of exact two-dimensional equilibria, is studied. The examined model reproduces the current sheet bending and shifting in the vertical plane, arising from the Earth dipole tilting and the solar wind non-radial propagation. The generalized model allows magnetic configurations with equatorial magnetic field decreasing in tailward direction as slow as $1/x$, contrary to the original Kan model ($1/x^3$); magnetic configurations with a single X-point are also available. The analytical solution is compared with the empirical T96 model in terms of the magnetic flux tube volume. It is found that parameters of the analytical model may be adjusted to fit a wide range of averaged magnetotail configurations. The best agreement between analytical and empirical models is obtained for the midtail at distances beyond $10-15\,R_E$ at high levels of magnetospheric activity. The essential model parameters (current sheet scale, current density) are compared to Cluster data of magnetotail crossings. The best match of parameters is found for single-peaked current sheets with medium values of number density, proton temperature and drift velocity.

## 1 Introduction

Studies of magnetosphere dynamics, including substorm events, require a relevant current sheet (CS) stability analysis. This in turn requires a proper choice of the background magnetoplasma configuration. In applications to collisionless plasma, the background equilibrium is to be derived from a solution of the kinetic Vlasov-Maxwell equations. A number of such solutions are derived both numerically (e.g., Burkhart et al., 1992; Pritchett and Coroniti, 1992; Cargill et al., 1994, and others), and analytically (e.g., Schindler and Birn, 2002; Yoon and Lui, 2005; Sitnov and Merkin, 2016; Vinogradov et al., 2016). All these solutions describe symmetric planar current sheets, the only approximate equilibrium solution for bent CS was introduced in the paper of Panov et al. (2012), where the authors present an analysis of direct THEMIS and GOES observations of plasma sheet evolution near substorm onset. Panov et al. (2012) have found the CS bending to be a source of the tailward growing normal magnetic field component $B_z$ (in the present paper we use the reference system with $x$ axis pointing tailward, $y$ axis pointing dawnward, and $z$ axis pointing north). Hence, bending of the current sheet turns out to be an important parame-

ter for the sheet stability, controlled by the sign of the derivative $\partial B_z/\partial x$ (e.g., Hau et al., 1989; Erkaev et al., 2007, 2009; Pritchett and Coroniti, 2010) in many instances.

This result is in line with previous findings revealing that the configuration asymmetry can be an important factor of magnetosphere dynamics. Particularly, Kivelson and Hughes (1990) have first suggested that the CS bending may drop down the reconnection onset threshold. This idea was confirmed later, when Partamies et al. (2009) have noticed the seasonal variations in the number of substorm events with maximums in winter and summer periods, when dipole tilt angle is bigger (the known geomagnetic activity maximums, e.g., in Kp index, are registered contrary around the equinoxes).

Later, this effect was investigated in details in the paper of Kubyshkina et al. (2015), where it was shown that the substorm probability is higher for about $10 - 25\%$ during the periods with tilt angle $> 15°$, as compared to the periods with smaller tilt angles. The direction of the solar wind (SW) flow also affects the substorm probability, it grows for $10 - 20\%$ when SW flow direction enforces the CS tilt to encrease. The statistical analysis has shown that the average substorm intensity (defined by AL value during the event) is lower for larger effective tilts (dipole tilt angle plus solar wind flow inclination). In other words, a large number of weak substorms occur in those time intervals when effective tilt angles are high, and less number of more intense substorms is observed when tilt angles are small. This agrees also with the results of Nowada et al. (2009) study, where both AL and AU indices were analyzed for the intervals of negative interplanetary magnetic field $B_z$.

In the same paper of Kubyshkina et al. (2015), the dependence of magnetotail lobe magnetic field (as a proxy of the magnetic flux) on the dipole tilt angle was studied by means of empirical modeling. The average lobe field was found to be smaller for all radial distances in a case of non-zero tilt angles. The decrease reached $10 - 20\%$ for maximum tilt angle. This result is reasonable under the assumption that substorm onsets require a lower energy input during the periods of increased dipole tilt. Next, in the paper of Semenov et al. (2015) it was found that there is a clear dependence of the substorm probability on the jumps of the $z$ component of the SW velocity (asymmetric factor), while the jumps of number density or plasma pressure (symmetric factor) turn out to be noneffective. At last, we should note that the Earth's dipole tilt angle undergoes daily and seasonal variations in the interval of about $\pm 35°$, so that it is equal to zero twice a day within about 4 months a year, and during other 8 months it is never zero. In addition, the solar wind flow direction varies for about $\pm 6°$. These variations produce CS inclination, bending and shift from the ecliptic plane. Therefore, the simplest solar wind-magnetosphere configuration (vertical dipole, planar CS, radial solar wind) adopted by the majority of models, is rather untypical and the development of the relevant bent CS models is highly-demanded.

The first exact solution for two-dimensional (2D) equilibrium bent CS with non-zero dipole tilt was presented in short notes of Semenov et al. (2015). This solution generalizes the well-known Harris-Fadeev-Kan-Manankova equilibria family (see Yoon and Lui (2005)). In the present paper we investigate the obtained solution to estimate its relevance for the magnetotail CS modeling and stability analysis. For this end, we compare the analytical solution with the empirical Tsyganenko (1995) T96 model and define the analytical model parameters, providing the best agreement.

The paper is organized as follows. In section 2 we describe the analytical solution for bent CS. In section 3 we compare analytical and empirical T96 solutions. In section 4 we present the further generalization of the analytical model, providing

more realistic profiles of $B_z$ in the equatorial plane. Then, the model typical scales are compared with *in-situ* data. Discussion and conclusions finalize the paper in section 5.

## 2   Analytical solution

For two-component (proton+electron) isothermal plasma with Maxwellian distribution functions and constant current velocity the system of Vlasov-Maxwell equations can be reduced to the 2D Grad-Shafranov equation (see Schindler (1972); Yoon and Lui (2005)) for the dimensionless magnetic potential $\mathbf{\Psi} = (0, \Psi, 0)$,

$$\frac{\partial^2 \Psi}{\partial x^2} + \frac{\partial^2 \Psi}{\partial z^2} = e^{-2\Psi}. \tag{1}$$

The quantity $\Psi$ is normalized for $(-B_0 L)$, where $L = 2cT_i/(eB_0 V_i)$ is the typical scale of CS in the normal direction and $B_0 = \sqrt{8\pi n_0 (T_e + T_i)}$ is the lobe magnetic field, $n_0 = n_{0e} = n_{0i}$ is the typical number density, $T_{e,i}$ are the electron and ion temperatures, respectively, and $V_{e,i}$ are the corresponding drift velocities, fulfilling the condition

$$V_i/T_i + V_e/T_e = 0. \tag{2}$$

Eq. (2) expresses the condition of the zero electrostatic potential. The model of an ion-dominated CS, where $|V_i/V_e| > T_i/T_e$ is considered in the paper of Yoon and Lui (2004). In the case of Maxwellian distribution functions condition (2) can be satisfied by means of the proper choice of the reference system, while in the general case of non-Maxwellian distribution functions it cannot be fulfilled (Schindler and Birn (2002)).

A series of analytical solutions of Eq. (1) was found by Walker (1915), shown that the solution may be expressed via an arbitrary generating function $g$ of the complex variable $\zeta = x + iz$,

$$e^{-2\Psi} = \frac{4|g'|^2}{(1+|g|^2)^2}, \quad g' = \frac{dg(\zeta)}{d\zeta}. \tag{3}$$

With the solution (3), the equilibrium magnetoplasma configuration takes the form

$$\Psi = \ln\left(\frac{1+|g|^2}{2|g'|}\right), \tag{4}$$

$$n = \exp(-2\Psi), \qquad p = 0.5\exp(-2\Psi), \tag{5}$$

where $p$ is the plasma pressure. By definition, the dimensionless magnetic field components are $B_x = -\partial\Psi/\partial z$ and $B_z = +\partial\Psi/\partial x$.

The particular choice of the generating function $g$ specifies the particular CS model. In the current paper we consider the family of Harris-like models, including the classical Harris (1962) current sheet, the Fadeev et al. (1965) solution (Harris sheet complemented by an infinite chain of magnetic islands along the neutral plane), the Kan (1973) solution (Harris sheet with quasi-dipole), and the Manankova et al. (2000) solution, representing the combination of all previous models. The last one is specified by the generating function

$$g(\zeta) = f + \sqrt{1+f^2}\exp\left[i\left(\zeta - \frac{b}{\zeta - a}\right)\right]. \tag{6}$$

Solution (6) contains three real parameters $a$, $b$ and $f$, where $a$ specifies the shift along the $x$ axis, $b$ controls the field line elongation, and $f$ defines the current density in the magnetic islands. Generating functions for other listed models are the special cases of the function (6). Namely, one should set $(f = 0, a \neq 0, b \neq 0)$ for the Kan solution; $(f \neq 0, a = b = 0)$ for the Fadeev solution; and $(f = a = b = 0)$ for the Harris solution.

The solution for a bent CS is developed in the paper of Semenov et al. (2015) by substituting the complex parameters $a \rightarrow ia$ and $b \rightarrow b_0 e^{i\varphi}$ in Eq. (6). The complex parameter $a$ controls the shift of the CS in the $z$ direction and $\varphi$ controls the dipole tilt angle. For the case of bent CS without plasmoids (Kan-like model, $f = 0$) the solution (6) takes relatively simple form,

$$\Psi = \ln\left(\frac{\cosh Z_*}{\sqrt{W}}\right), \tag{7}$$

$$Z_* = z - \frac{b_0 x \sin(\varphi) - b_0(z - a)\cos(\varphi)}{R^2}, \tag{8}$$

$$W = 1 + \frac{b_0^2 + 2b_0(x^2 - (z-a)^2)\cos(\varphi) + 4b_0 x(z-a)\sin(\varphi)}{R^4}, \tag{9}$$

where $R^2 = x^2 + (z - a)^2$. Configurations of this type possess a dipole singularity at $(x, z) = (0, a)$ and two additional singularities at $(x, z) = (\pm\sqrt{b_0}\sin(\varphi/2), a \mp \sqrt{b_0}\cos(\varphi/2))$, rotating twice as slow as a dipole does. Hence, the effective dipole tilt is equal to $\varphi/2$. For positive tilt angles the CS is bent and uplifted over the ecliptic plane, and for negative tilts the CS is shifted down.

The set of magnetic configurations for dipole tilt angle PHI= $\{0, 30, 60, 120\}$ degrees clockwise (PHI= $-\varphi/2$) is shown in Fig. 1. The two first cases ($0°$ and $30°$) can be observed in the Earth's magnetosphere, and other cases are shown here to illustrate the model behavior. White asterisks on panels c) and d) of Fig. 1 mark the X-points ($B_x = B_z = 0$), being an attribute of the Kan-like solution. In the symmetric Kan model the X-point is located at infinity, but in bent sheets it starts to approach the dipole with increasing tilt angle. This X-point is not produced by magnetic reconnection, and it does not break a steady state equilibrium of the CS. On the other hand, the appearance of the X-point can be considered as a manifestation of potentially unstable configuration. In such a case, the X-point motion towards dipole with increasing tilt angle could mean that CS evolves toward an unstable state. According to the solution (7-9), the X-point location depends also on the CS width $L$ and model parameter $b_0$. The X-point position as a function of $\varphi$ is plotted in Fig. 2 for three values of $b_0$, corresponding to three different levels of geomagnetic activity (see Fig. 4, right column). It is seen that for tilt angles $|\varphi/2| < 45°$ the X-point stays very far beyond $60\,R_E$ for any realistic value of $L$ and $b_0$. E. g., for $b_0 = 8$ (the value, corresponding to quiet magnetotail) and $|\varphi/2| = 45°$ the X-point stays as far as $\approx 340\,L$. For $|\varphi/2| = 60°$ (almost two times more than the Earth maximal dipole tilt) an approach to $8.5\,L$ is achieved.

## 3  Comparison with the T96 model

Topologically, magnetic configurations plotted in Fig. 1 are very similar to that of the Earth's magnetosphere. However, to estimate the relevance of the analytical solution one should compare some important numerical characteristics of the CS model with the corresponding values registered in real observations. This can be done utilizing empirical magnetic field models,

providing realistic averaged magnetospheric configurations at various levels of magnetospheric activity. Of course, we should keep in mind that the real magnetosphere is an essentially tree-dimensional structure. Following the dipole tilt (and solar wind flow direction) variations, the magnetotail CS bends and shifts from the equatorial plane in $z$ direction (at most $\sim 3\,R_E$ for maximum tilt) and also warps in the $y$ direction. These effects are well-pronounced in empirical magnetospheric models, but the 2D analytical model is evidently unable to reproduce all these complex deformations. Therefore, we restrict our study to the noon-midnight plane $y = 0$, and the two main effects manifested in that plane: CS bending and shifting in $z$ direction.

To explore the appropriacy of the here presented analytical solution for bent CS, we compare the predicted magnetic flux tube volume (a proxy for the entropy) with that calculated from the empirical model of Tsyganenko (1995) T96. We consider the flux tube volume (FTV) instead of the entropy, since the analytical solution is isothermal. This quantity is chosen due to its importance for the magnetotail dynamics. As it was claimed by Birn et al. (2009) and verified by *in-situ* data analysis (Sergeev et al. (2014)), any bursty bulk flow (BBF), produced by reconnection in the magnetotail and moving toward the Earth, stops near that particular point, where the entropy of the ambient plasma is equal to that inside the BBF. The distribution of entropy along the magnetotail is also an important factor for the stability analysis (Birn et al. (2009)) and for the study of wave (oscillations) generation and dissipation (Panov et al. (2016)).

The FTV is determined in the same way for both analytical and empirical models: we integrate $dS/B$ along magnetic field lines, where $dS$ is the field line length element and $B = \sqrt{B_x^2 + B_z^2}$. In the T96 model the location of the flux tube is computed by means of field line tracing, in the analytical model this is a curve of constant $\Psi$. As a first step, the values of FTV of a single flux tube are compared. The model parameters correspond to the quiet magnetospheric conditions with tilt angle of $30°$ clockwise (see the legend of Fig. 4b). FTVs are calculated along the magnetic field line with a node at $(x, z) = (30, -2.4)$. To eliminate singularities, we excluded the near-Earth region $x < 5\,R_E$, so that the total FTVs are calculated as $\int_{(30, -2.4)}^{(5, 1.5)} dS/B$. The results are shown in Fig. 3 by the blue curve for T96 and by red curve for the Kan model. The values of FTV, normalized for total FTV, are plotted as a function of $x$. It is seen that two models demonstrate rather close results. $90\%$ of FTV are provided by the farther half of the tube, $x \in [15, 30]\,R_E$, and $50\%$ of FTV are concentrated in the most distant interval within $3 - 4\,R_E$ in the $x$ direction from the tube node.

Then, FTVs, calculated by means of analytical and empirical models, are compared at different levels of magnetospheric activity, characterized by input parameters of the T96 model (Dst index, the SW dynamical pressure, $p_{dyn}$, and SW magnetic field components $B_y^{sw}$ and $B_z^{sw}$). Three sets of parameters are taken to specify the quiet magnetotail $\{Dst = -10, p_{dyn} = 2\,\text{nPa}, B_z^{sw} = 2\,\text{nT}\}$, substorm conditions $\{Dst = -50, p_{dyn} = 3\,\text{nPa}, B_z^{sw} = -3\,\text{nT}\}$, and storm $\{Dst = -150, p_{dyn} = 6\,\text{nPa}, B_z^{sw} = -7\,\text{nT}\}$. Magnetic field component $B_y^{sw}$ was set equal to zero. Parameters $a$ and $b_0$ of the analytical solution (7–9) are found numerically to minimize the standard deviation between two models. The results (FTV versus $x$ coordinate of the flux tube node) are presented in Fig. 4, where red lines plot analytical solutions and black ones plot the T96 results. The left column shows the symmetrical case (zero dipole tilt), and the right column corresponds to the dipole tilt angle of $30°$ clockwise.

One can see that the agreement between two models is quite good with the maximal standard deviations varying within $2 - 11\%$. The values of $d_{max} = \sigma_{max}/\langle FTV \rangle$ are given in legends of Fig. 4, where $\sigma$ is the standard deviation and $\langle FTV \rangle$ is the average FTV. The better agreement is achieved for disturbed magnetospheric conditions, i.e. the analytical model describes

the stretched CSs even better than the thicker ones. It is found that minimal difference between two models is obtained when parameter $a$ is very close to the medium neutral sheet position determined from the empirical model. The best-fit value of the parameter $b_0$, controlling the field lines stretching and the CS thinning, depends on the level of activity and the dipole tilt angle. It grows from $8.8$ for the quiet magnetosphere to $51$ for storm conditions. At any fixed distance, the stretching of field lines makes the FTV to decrease with growing magnetospheric activity. E.g., at the distance of $x = 30\,R_E$ it changes from $\approx 5\,R_E/nT$ for "quiet" conditions to $\approx 3\,R_E/nT$ for "substorm" conditions and to the $\approx 1.6\,R_E/nT$ for "storm-time" conditions. Contrary, the asymmetric deformation of CS (dipole tilt angle) enforces the FTV to increase.

Fig. 4 shows a comparison of two models within the large interval $x \in [5, 30]$. To detect the best-match region we performed the same analysis for eight short overlapping intervals $x \in [7.5, 12.5] + 2.5n$, where $n = 0, 1, ..., 8$. The normalized standard deviation as a function of $x_{0n}$, where $x_{0n}$ is the center of corresponding interval, is shown in Fig. 5, where three features are observed: (1) standard deviation grows toward the Earth and exceeds $10\%$ for $x < 15\,R_E$ for all activity levels; (2) deviations are bigger for the more quiet magnetosphere environment; (3) deviations are smaller for a tilt angle of $30°$. As compared to results of the large interval analysis (Fig. 4), dependence on the activity level is the same, and dependence on the tilt angle demonstrates opposite behavior. Overall, analytical and empirical models show good agreement beyond $15\,R_E$, improving with growing activity.

## 4   Normal magnetic component and current density

The results of the previous section show that parameters of the asymmetric Kan-like model may be adapted to provide rather good agreement with the magnetotail CS, especially in a distant tail beyond $15 - 20\,R_E$, and especially for bent current sheets. However, until now the practical usage of this model encountered the substantial obstacle, related to the behavior of the normal magnetic field component. It can be easily checked that in the distant tail the Kan model yields $B_z \sim 1/x^3$, while in reality $B_z$ decreases as $1/x$ or even slower (e.g., Behannon and Ness (1966); Mihalov et al. (1968); Behannon (1970); Wang and Lyons (2004); Yue et al. (2013)). For plane and axially-symmetric current sheets the solution with $B_z \sim 1/x^\alpha$ with arbitrary $\alpha$ is found in Vasko et al. (2013). For Kan-like models considered in the current paper the $B_z$-problem may be solved by introducing one more parameter in the generating function $g(\zeta)$. With the additional parameter $n$, general asymmetric model takes the form (compare to Eq. 16 of Yoon and Lui (2005))

$$g(\zeta) = f + \sqrt{1 + f^2}\,\exp\left[i\left(\zeta^n - \frac{b}{(\zeta - a)^k}\right)\right]. \tag{10}$$

Assuming $\{f, n, k\}$ to be real values, $a = a_1 + ia_2$, and $b = b_0 \exp(i\varphi)$, we derive

$$\Psi = \ln\left(\frac{f\cos X_* + \sqrt{1+f^2}\cosh Z_*}{\sqrt{W}}\right), \tag{11}$$

$$X_* = r^n \cos(n\vartheta) - \frac{b_0}{R^k}\cos(k\Theta - \varphi), \tag{12}$$

$$Z_* = r^n \sin(n\vartheta) + \frac{b_0}{R^k}\sin(k\Theta - \varphi), \tag{13}$$

$$W = n^2 r^{2(n-1)} + \frac{b_0^2 k^2}{R^{2(k+1)}} + 2nkb_0\frac{r^{n-1}}{R^{k+1}}\cos\left[(n-1)\vartheta + (k+1)\Theta - \varphi\right], \tag{14}$$

$$r = \sqrt{x^2 + z^2}, \qquad\qquad \vartheta = \arctan\left(\frac{z}{x}\right), \tag{15}$$

$$R = \sqrt{(x-a_1)^2 + (z-a_2)^2}, \qquad \theta = \arctan\left(\frac{z-a_2}{x-a_1}\right). \tag{16}$$

For symmetric Kan-like CS without plasmoids ($a = 0$, $f = 0$, $\varphi = 0$), the quantity $B_z$ at the $x$ axis takes the simple form $B_z(x,0) = -(\partial W/\partial x)/(2W)$. It is seen that the Kan solution ($n = 1$) is the only degenerated case when the first term of $W$ turns to 1 and its derivative to zero, hence in the distant tail $B_z \sim (1/x^{2+k})$ due to the rightmost term of expression (14). For any $n \neq 1$ we have $(\partial W/\partial x)/W \rightarrow O(1/x)$.

Parameter $n$ controls flaring of magnetic field lines; values of $n > 1$ enforce strong convergence of the CS field lines toward the $x$ axis, hence location of the X-line is drastically dependent on $n$. This feature is illustrated in Fig. 6, where four symmetric magnetic configurations with ($a = 0$, $\varphi = 0$, $f = 0$, $b_0 = 1$, $k = 1$) and $n = \{0.95, 1, 1.05, 1.1\}$ are plotted. In Fig. 7 reverse values of the equatorial magnetic field, $B_z^{-1}(x,0)$, are plotted for several sets of the model parameters. The set of green curves illustrates contribution of the parameter $k$. The set of violet curves shows the effect of the parameter $b_0$ variation. The set of solid curves demonstrates the parameters $n$ impact. It is seen that: a) all curves except the red one (original Kan solution, $n = 1$) tend to $O(x)$, b) numerical values of $B_z$ are highly variable depending on different combinations of parameters $\{b_0, k, n\}$.

In two dimensions, contributions of parameters $n$ and $\varphi$ are shown in the next two plots. Figs. 8 and 9 present $J_y(x,z)$ and $B_z(x,z)$, respectively, for six sets of the model parameters, where parameters $a_1 = 0$, $a_2 = -0.03$, $f = 0$, $b_0 = 22.13$, $k = 1$ are the same. Panels a) show the solutions for ($n = 0.995$, $\varphi = 0$). Panels b) show the solutions for the bent sheet ($n = 0.995$, $\varphi/2 = 30$). On panels c) solutions for a plane substorm CS model (see Fig. 4c) with $n = 1$ and $\varphi = 0$ are shown; the bent sheet ($n = 1$, $\varphi/2 = 30$) quantities are plotted on panels d). On panels e) parameter $n = 1.005$ and $\varphi = 0$, and on panels f) $n = 1.005$, $\varphi/2 = 30$.

Fig. 8 demonstrates that the CS width is almost uniform on $x$ and is not affected by tilt angle, controlling only the sheet location (vertical shift may be recouped by the proper choice of parameter $a_2$). With increasing parameter $n$, the sheet is thinning and, correspondingly, the peaking current density is growing. The same effect is produced by enhanced geomagnetic activity. Comparison of current densities for quiet and storm conditions (not shown) reveal 20% reduction of the CS width and 20% grow of the peaking current density.

Fig. 9 shows that even so weak variation of parameter $n$ affects the distribution of $B_z$, mostly near to the sheet center. The range of appropriate values of $n$ is restricted from above by the solution geometry (X-point location). Say, for current

model parameters and with $n = 1.01$, the X-point is located at $x \approx 65$ in a plane sheet, and it approaches $x \approx 50$ for $\varphi/2 = 30$. Expectably, the increase of tilt angle $\varphi$ enhances the value of $B_z$, so that for $\varphi/2 = 30° $ $B_z$ is growing 10 times.

The solution (11–16) is written in normalized units, where the magnetic field is normalized for the lobe value $B_0$, and normalization constants for the length scale and current density are

$$5 \quad L \;=\; \frac{2cT_i}{eB_0V_i} = 2\cdot 10^3 \cdot \frac{T_i\,[keV]}{B_0\,[nT]\,V_i\,[km/s]},\; 10^3\,km \tag{17}$$

$$J_0 \;=\; \frac{cB_0}{4\pi L} = 0.8\cdot \frac{B_0\,[nT]}{L\,[10^3\,km]},\; nA/m^2 \tag{18}$$

To estimate the relevance of this scaling, we make use of Cluster data of magnetotail CS crossings, presented in Table 1 of Runov et al. (2006). Assuming $V_i = \sqrt{V_x^2 + V_y^2 + V_z^2}$ and $B_0 = B_L$, the quantities $L$ and $J_0$ are calculated. The plot of $J_0(L)$ is shown in Fig. 10. Most of the points, that we call "regular", lie within the interval of $L \in [2, 8]\,10^3\,km$ and $10 \quad J_0 \in [3, 15]\,nA/m^2$ (red asterisks). Other points represent extremely small values of CS parameters, such as very low ion temperature ($T_i < 2\,keV$, blue crosses), drift velocity ($V_i < 35\,km/s$, blue diamonds), and number density ($n_i < 0.2\,cm^{-3}$, blue asterisks). A single case of extremely high value of $V_i = 659\,km/s$ is marked by a magenta circle.

Fig. 11 shows the model normalization constant $J_0$ versus peaking observed perpendicular current density (blue curve on Fig. 2 of Runov et al. (2006)). It is seen that analytical estimate and measured values of $J_0$ mismatch in all extreme cases of Fig. 10. In other cases ("regular" points, red asterisks) the model estimate agrees with observed values with an accuracy up to a coefficient $k \in [0.5, 2]$ (except for the cases 4 and 23 of Runov et al. (2006), when discrepancy reaches 2.5 times). Thus, the best match of current densities is found for cases $\{1 - 3, 5 - 7, 11, 15 - 18, 22, 25 - 28\}$, which are mostly single-peaked current sheets. The analytical model (11–16) preserves basic features of the initial Harris solution, hence it is unable to resolve the complex CS structure, such as bifurcated or embedded current sheets (see, e.g., Hoshino et al. (1996); Nakamura et al. (2006); Runov et al. (2006); Artemyev et al. (2009); Petrukovich et al. (2015)). It means that the cross-sheet profiles of current density in our model (not shown) resemble the Harris profiles, shown on Figs. 2 and 3 of Runov et al. (2006). Hence, analytical estimate of the CS width exceeds usually the real values. However, in some cases (e.g., cases 3, 27 and 28) the Harris profiles may be more or less relevant to real current sheets.

## 5   Discussion and conclusions

In empirical models (T89, T96, T01, TS05, etc.) magnetic field configurations with any plasma populations are not force-balanced since $\nabla \times [\mathbf{j} \times \mathbf{B}] \neq 0$, or there is no $\nabla P$ to balance Ampere's force (Zaharia et al. (2003)). That is why we crucially need kinetic force-balanced CS models for many magnetospheric studies, such as wave generation in plasma, CS stability analysis, and numerical simulations of magnetotail dynamics. So far these studies were restricted by purely symmetric background equilibria. In this paper we present the extension of the well-known family of exact kinetic Harris-Fadeev-Kan-Manankova solutions to the 2D bent CS. This extension is really important, since the Earth dipole is tilted most of the time.

To validate the obtained analytic solution for bent CS we performed a comparison with T96 model, used as a proxy of realistic averaged magnetospheric configuration. It is shown that the proposed model provides a reasonable approximation for

the magnetotail CS in a wide range of dipole tilt angles and geomagnetic activity levels. Particularly, the parameters of the analytical model can ever be adjusted to fit the behavior of the magnetic FTV with an accuracy of about 10% for all distances from 5 to 30 $R_E$ tailward. For short segments ($5\,R_E$) of the CS, located beyond $15\,R_E$, the agreement may be improved up to 5% (except the case of the bent CS at quiet magnetospheric conditions). The agreement between analytical and empirical models is found to be better for the stretched magnetic configuration, i.e., for the pre-substorm conditions.

Notably, such a good agreement is obtained for the simplest three-parametric Kan-like model (7–9), where parameter $a$ controls the CS displacement from the equatorial plane, parameter $b_0$ controls magnetic field lines stretching, and parameter $\varphi$ specifies the CS bending. For further studies the more general model (11–16) can be considered, where additional parameters $n$ and $k$ provide the more accurate adjustment of the magnetoplasma quantities. More over, for sub-Alfvénic plasma, i.e., for the low-activity periods, all model parameters may be treated as time-dependent quantities (Wolf (1983); Semenov et al. (2015)). The time-dependent approach in such a modeling is not appropriate for the periods of explosive activity, such as storms and substorms, when BBFs with Alfvénic speed are produced.

Of course, the suggested analytical model is still far from universality. One significant limitation of this model is related to the isothermal constraint. This constraint may be released for four-component (two positive + two negative) plasma with bi-Maxwellian distribution functions for each particle specie (Kan (1973); Voronina and Kan (1993)). In such a case the condition (2) takes the form $V_{ik}/T_{ik} + V_{ek}/T_{ek} = 0$, where $k = \{1, 2\}$. If two plasma components give zero contribution in the current velocity, $V_{i2} - V_{e2} = 0$, the Eq. (1) stays valid for nonuniform plasma temperature (Voronina and Kan (1993)). The four-component-plasma model could be probably appropriate for magnetotail studies at high levels of geomagnetic activity. Indeed, in the quiet magnetotail the population of ions $\{O^+, O^{++}, He^{++}\}$, penetrating from the ionosphere, is less than 1% (Lennartsson et al. (1986)), hence the approximation of "proton+electron" plasma is relevant. With the growth of geomagnetic activity, the $O^+$ contribution becomes essential during the main and recovery phases of intensive storm events. However, practical application of the non-isothermal model requires thorough studies, going beyond the scope of the present paper.

The constancy of the proton temperature is not reflected in observations (e.g., Kissinger et al., 2012; Wang et al., 2012), hence the isothermal model may be considered as a first approximation only. Though, for some local analysis it seems to be rather suitable due to the small ($\sim 10 - 20\%$) cross-cut variations of proton temperature, detected in observations of central-peaked current sheets (see Fig. 5 in Runov et al. (2006)). In such sheets, inaccuracy of the constant-temperature estimate does not exceed the model inaccuracy in current density or CS width.

Other model limitations are the two-dimensionality, and isotropy of the plasma pressure. Even with these limitations, the model stays appropriate for the wide class of problems, mentioned in the beginning of the current section. Particularly, we lay hopes that application of the presented model can stimulate investigations on the magnetotail CS stability to resolve the question suggested by Kivelson and Hughes (1990): why symmetric CS can accumulate magnetic flux energy more effective, and does the threshold of substorm-initiating instability depend on degree of the CS bending.

In summary:

– An exact 2D bent CS equilibrium, built by means of generalization of the Harris-Fadeev-Kan-Manankova family of symmetric solutions of the Vlasov-Maxwell equations, is considered. The examined model reproduces the effects, related

to the Earth dipole tilt and CS bending. The further generalization releases degeneracy of the original model, which yielded too fast decrease of the normal magnetic component.

– Parameters of the asymmetric model may be adjusted to reproduce the realistic distribution of the magnetic flux tube volume at any level of geomagnetic activity; with enhancing activity the model relevance improves. The model typical scales for CS width and current density match the corresponding parameters of the *in-situ* registered single-peaked current sheets with medium values of number density, proton temperature and drift velocity; disagreement does not exceed factor 2.

– The asymmetric solution does not contain any limitation for the tilt angle values, hence the model is appropriate for any Earth-like magnetosphere with arbitrary dipole inclination.

– The obtained bent CS solution contains the X-point, moving from infinity toward the dipole with the dipole tilt increase, staying still far beyond the lunar orbit for the Earth magnetotail realistic tilt angles. Much more effectively the location of the X-point is controlled by the new parameter $n$ of the generalized model (11–16).

*Acknowledgements.* This study has been supported by the Austrian Science Fund (FWF): P 27012-N27 and I 3506-N27, and by Russian Science Foundation (RSF) grant No 18-47-05001. D. B. K. thanks A. V. Egorova for her help with preparation of the pictures. The authors thank the reviewers for their help in improving the manuscript.

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

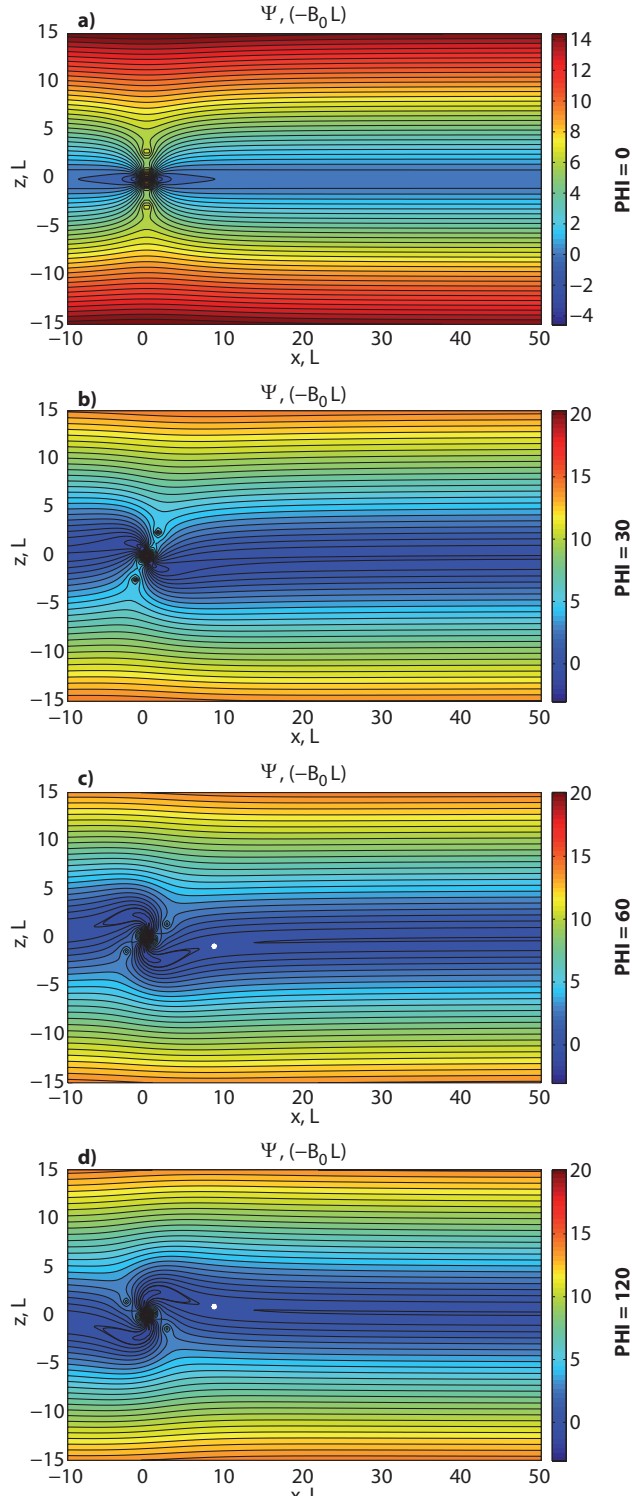

**Figure 1.** Magnetic potential $\Psi(x, z)$, calculated from the asymmetric Kan model (7–9) with parameters $a = 0$ and $b_0 = 8$. Solutions with dipole tilt angles PHI= $\{0°, 30°, 60°, 120°\}$ clockwise are plotted on panels {a, b, c, d}, respectively. PHI= $-\varphi/2$ of the analytical model. Spatial units are normalized for typical CS width $L$. Magnetic potential is normalized for $(-B_0 L)$. X-points are marked white.

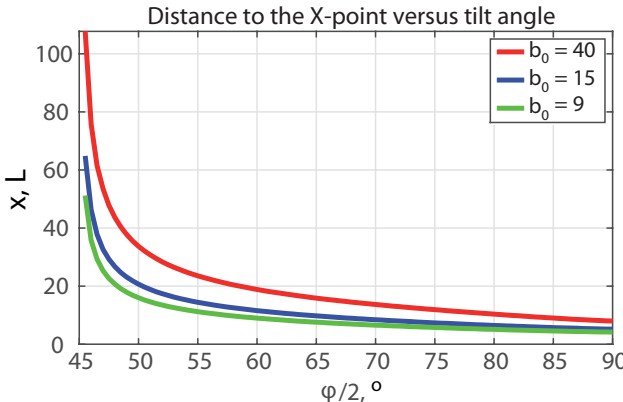

**Figure 2.** X-point location versus the effective tilt angle $\varphi/2$ in the asymmetric Kan solution (7–9) with $a = 0$ and parameter $b_0 = 40$ (red), $b_0 = 15$ (blue), and $b_0 = 9$ (green).

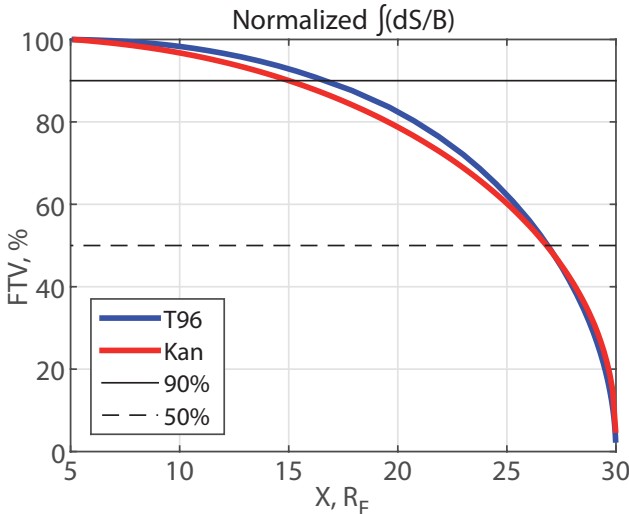

**Figure 3.** Flux tube volume $\int_{(30,-2.4)}^{(x,z)}(dS/B)$ normalized to the full FTV $\int_{(30,-2.4)}^{(5,1.5)}(dS/B)$, in percents, calculated by T96 model (blue) and by the Kan model (red) for the quiet conditions with tilt angle PHI= $30°$ clockwise (model parameter $\varphi = -60$). Other model parameters are given in the legend of Fig. 4b. The Earth is on the left.

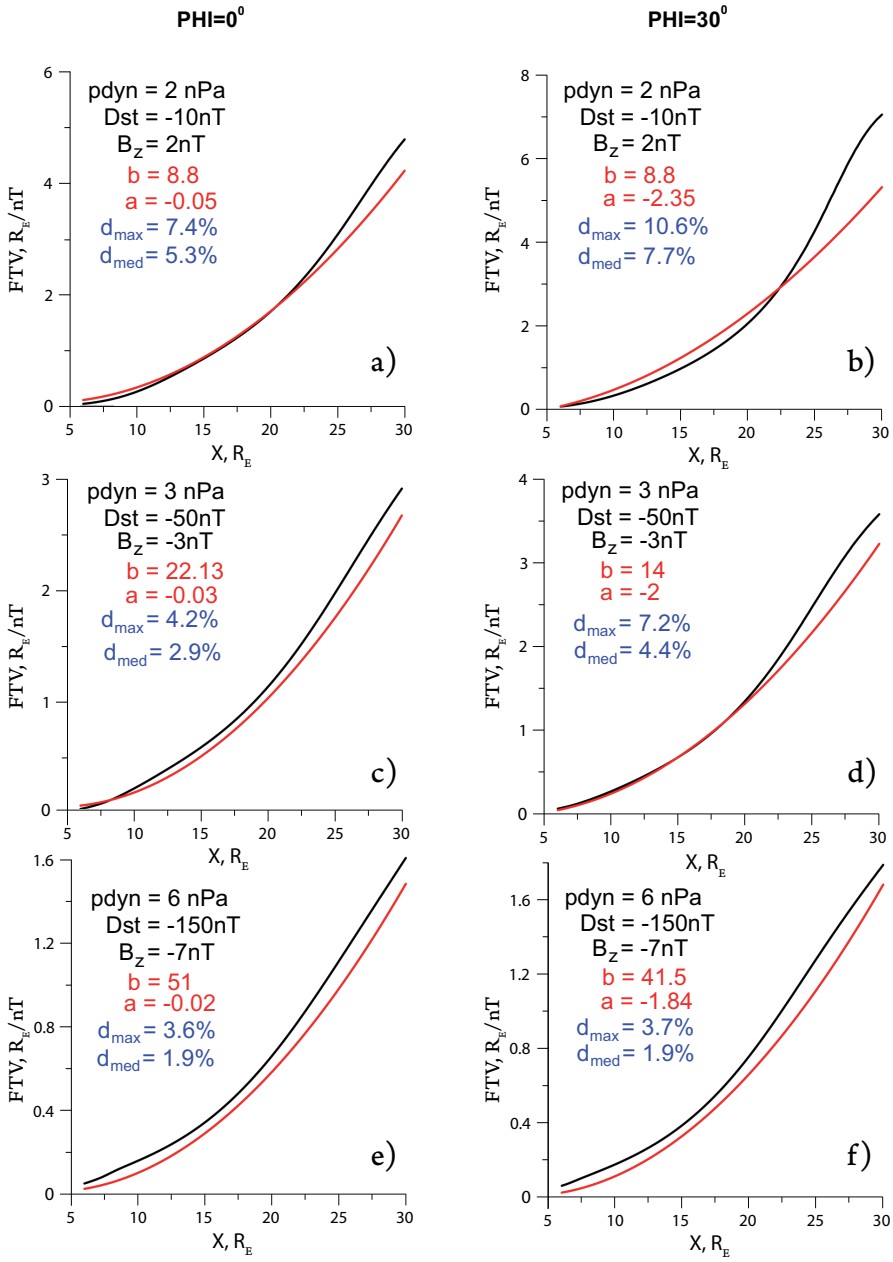

**Figure 4.** Flux tube volumes: analytical solution (red curves) and T96 (black curves) for quiet (top row), substorm (middle row) and storm (bottom row) conditions are plotted for tilt angles $0°$ (left column) and PHI= $30°$ clockwise (right column). Input parameters for the T96 model (black text), for the Kan-like model (red text), and standard deviations normalized for average FTV (blue text) are given in legends. The Earth is on the left.

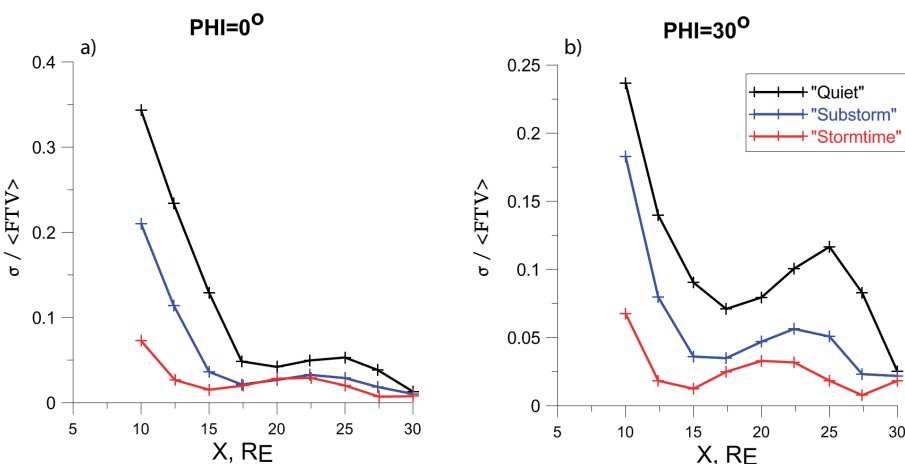

**Figure 5.** Comparison of analytical and empirical models for symmetric (dipole tilt $0°$, on the left) and bent ($30°$ clockwise, on the right) current sheets. Standard deviations, $\sigma$, normalized for average FTV, are shown as functions of $x_0$, where $x_0$ is the center of the region under consideration $[x_0 - 2.5, x_0 + 2.5]$, for quiet (black curves), substorm (blue curves) and storm (red curves) conditions. The Earth is on the left.

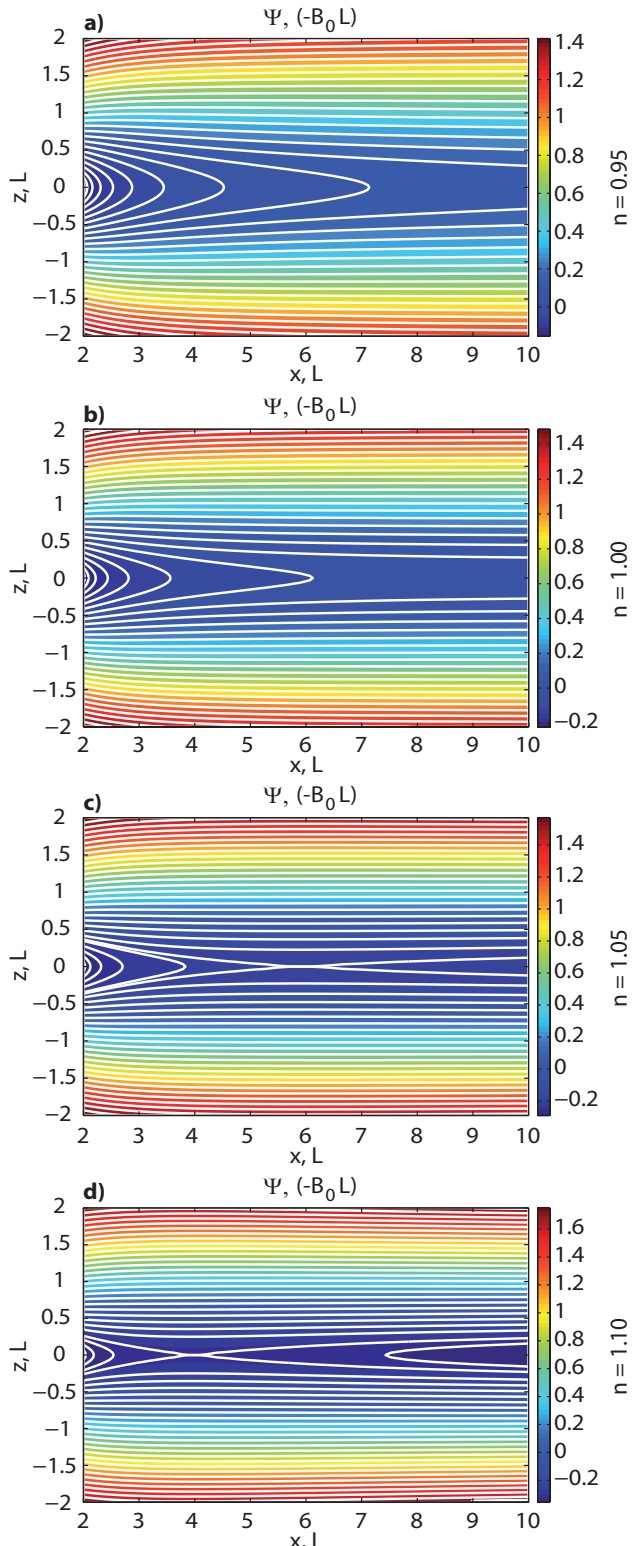

**Figure 6.** The values of magnetic potential $\Psi(x,z)$, calculated from analytical model (11–16), are shown by color for model parameter $n = \{0.95, 1, 1.05, 1.1\}$ on panels $\{a, b, c, d\}$, respectively. Other parameters $\{a = 0, b_0 = 1, \varphi = 0, f = 0, k = 1\}$ are the same. Magnetic field lines are plotted by white curves. Panel b) shows the original Kan solution.

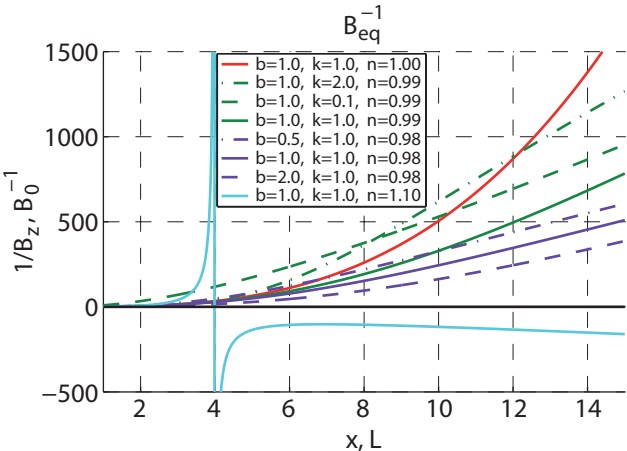

**Figure 7.** Profiles $1/B_z(x,0)$ for symmetric Kan-like CS, calculated from analytical model (11–16). Parameters $\{a_1 = 0, a_2 = 0, \varphi = 0, f = 0\}$ are the same. Other parameters are: $\{b_0 = 1, k = 1, n = 1\}$ (red), $\{b_0 = 1, k = 2, n = 0.99\}$ (dark-green dash-dotted), $\{b_0 = 1, k = 0.1, n = 0.99\}$ (dark-green dashed), $\{b_0 = 1, k = 1, n = 0.99\}$ (dark-green solid), $\{b_0 = 0.5, k = 1, n = 0.98\}$ (violet dash-dotted), $\{b_0 = 1, k = 1, n = 0.98\}$ (violet solid), $\{b_0 = 2, k = 1, n = 0.98\}$ (violet dashed), and $\{b_0 = 1, k = 1, n = 1.1\}$ (cyan). Red curve shows the original Kan solution. Units are normalized for CS typical width $L$ and for $B_0^{-1}$.

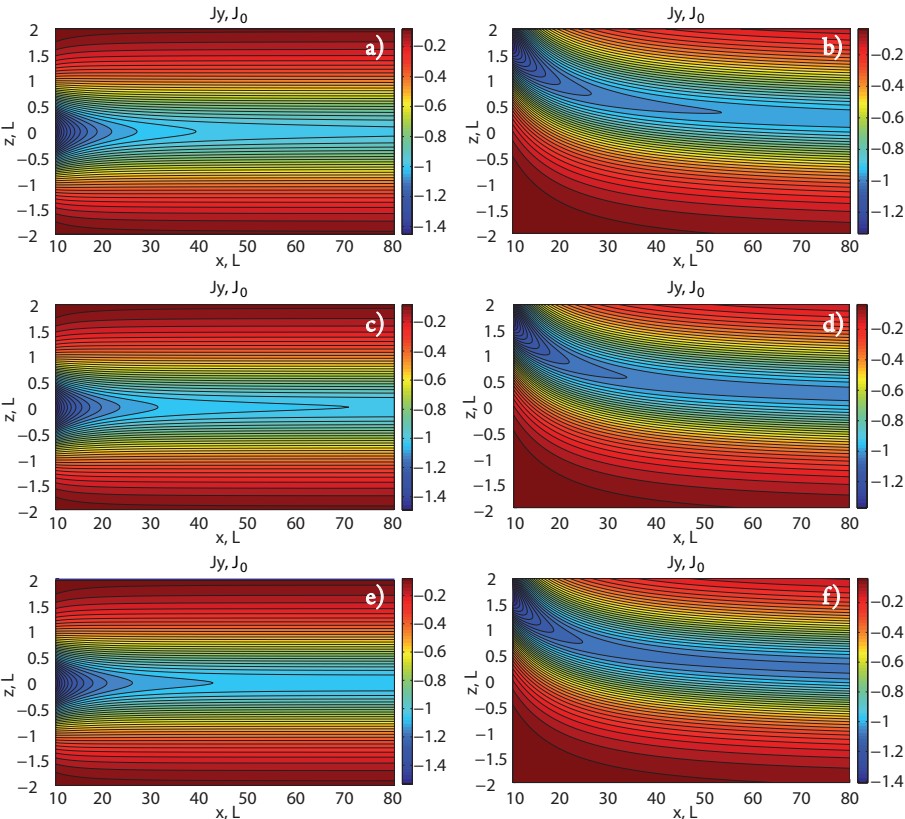

**Figure 8.** Current density $J_y(x, z)$ by analytical model (11–16) for $\{a_1 = 0, a_2 = -0.03, b = 22.13, f = 0, k = 1\}$ is shown for three values of the parameter $n$ for plane sheets (left column, $\varphi = 0$) and curved sheets (right column, $\varphi = 60$). Top row: $n = 0.995$. Middle row: $n = 1$. Bottom row: $n = 1.005$. Panel c) corresponds to the plane substorm sheet (see Fig. 4c). Units are normalized for CS typical width $L$ and $J_0 = cB_0/(4\pi L)$.

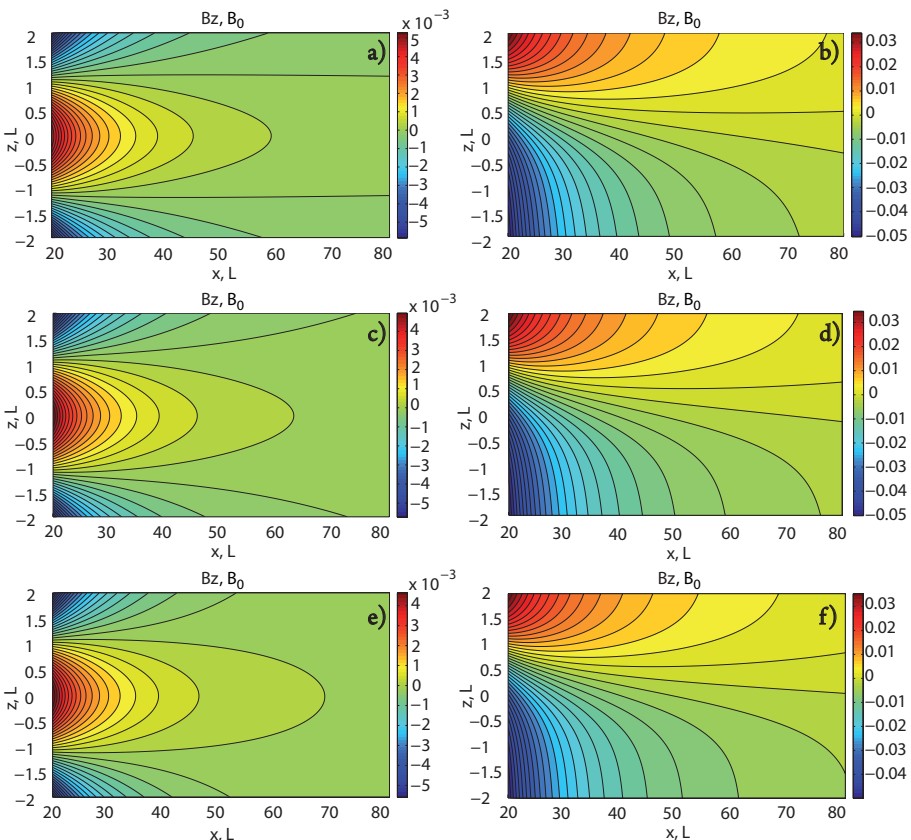

**Figure 9.** Magnetic field component $B_z(x,z)$ by analytical model (11–16) for $\{a_1 = 0,\, a_2 = -0.03,\, b = 22.13,\, f = 0,\, k = 1\}$ is shown for three values of the parameter $n$ for plane sheets (left column, $\varphi = 0$) and curved sheets (right column, $\varphi = 60$). Top row: $n = 0.995$. Middle row: $n = 1$. Bottom row: $n = 1.005$. Panel c) corresponds to the plane substorm sheet (see Fig. 4c). Units are normalized for CS typical width $L$ and $B_0$.

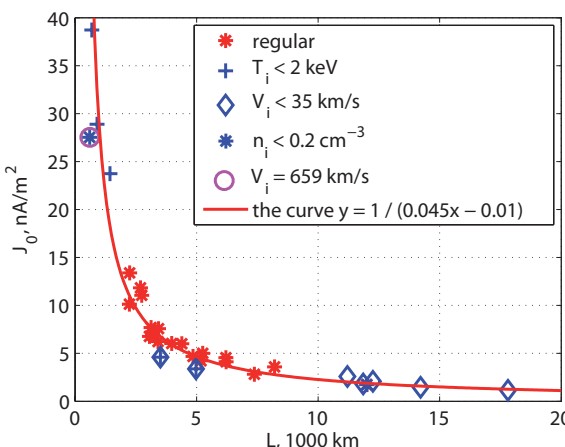

**Figure 10.** Analytical model units: current density $J_0$ vs. spatial scale $L$ from Eq. (17, 18) for Cluster data of current sheet crossings, presented in Table 1 of Runov et al. (2006). Blue crosses show cases of the lowest ion temperature, $T_i < 2\,keV$ (8, 9, 12, 13, 14 in Table 1 of Runov et al. (2006)), blue diamonds show cases of the lowest ion drift velocity, $V_i < 35\,km/s$ (8, 9, 10, 19, 21, 29, 30), and blue asterisks show cases (20, 24) of the lowest ion number density, $n_i < 0.2\,cm^{-3}$. Magenta circle shows the case 20 of extremely high velocity, $V_i = 659\,km/s$. All other "regular" cases are shown by red asterisks. The red line plots the fitting curve $y = 1/(0.045x - 0.01)$.

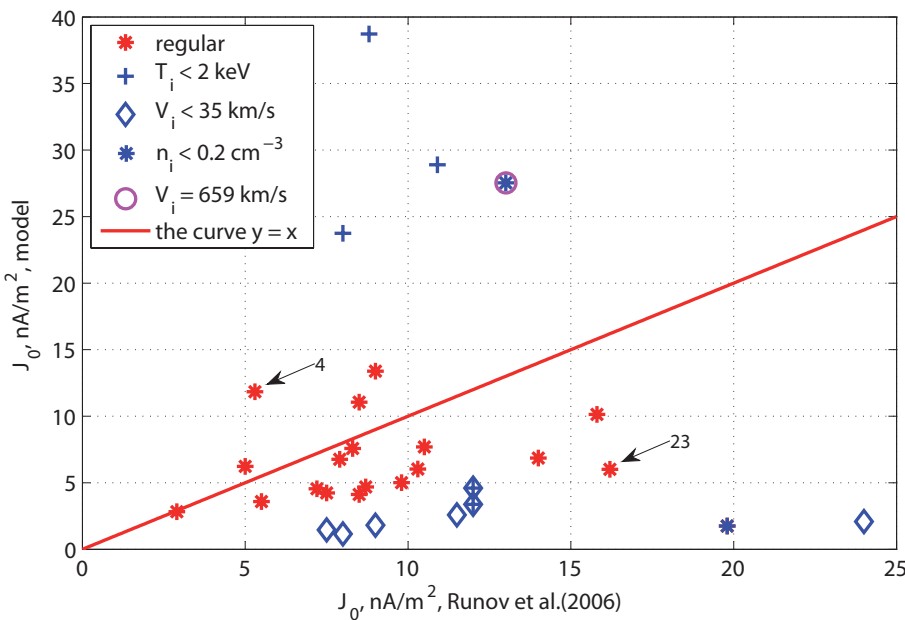

**Figure 11.** Typical current density $J_0$ of the analytical model vs. peaking perpendicular current density from Fig. 2 of Runov et al. (2006). Blue crosses show cases of the lowest ion temperature, $T_i < 2\,keV$ $(8, 9, 12, 13, 14$ in Table 1 of Runov et al. (2006)), blue diamonds show cases of the lowest ion drift velocity, $V_i < 35\,km/s$ $(8, 9, 10, 19, 21, 29, 30)$, and blue asterisks show cases $(20, 24)$ of the lowest ion number density, $n_i < 0.2\,cm^{-3}$. Magenta circle shows the case 20 of extremely high velocity, $V_i = 659\,km/s$. All other "regular" cases are shown by red asterisks. The red line plots $y = x$. Black arrows mark cases 4 and 23, demonstrating the largest (amongst red points) discrepancy of the observed and model values.