# Peer review of "On application of asymmetric Kan-like exact equilibria to the Earth magnetotail modeling"

_Annales Geophysicae, 2018_

## Referee Comment (RC1) · Anonymous Referee #1 · 12 Feb 2018

Review on paper "On application of asymmetric Kan-like exact equilibria. . ." submitted by Korovinskiy et al. This draft describes a new model of 2D magnetotail-like current sheet with the north-south asymmetry. Authors use this interesting generalization of the class of 2D models to simulate/mimic effects of the Earth's dipole inclination. The comparison of flux tube volumes calculated using the presented model and the empirical T96 model demonstrates the analytical model applicability. Paper is well written and logically consistent. It can be interested to AnGeo readers, but additional work should be done before publication.

My main concern corresponds to the model verification. . . using T96 is a good ap-

proach, because it allows to compare nonlocal model characteristics (i.e., flux tube volume). However, I would expect to see some comparison with actual spacecraft observations. Or, at least, some discussion of such comparison. Most of Authors are affiliated to IWF Graz, one of the centers of the magnetotail current sheet investigations. Thus, I'm surprised by absence of references to main important results about the magnetotail current sheets obtained in this Institute. Below Authors can find some recommendation about further model verification and list of small (mostly editorial) changes.

(1) There are several publications by Runov et al. (2005, 2006 AnGeo), Nakamura et al. (2006 SSR), Artemyev et al. (2008, 2009 AnGeo) and two reviews by Baumjohann et al. (2007, AnGeo) and Petrukovich et al. (2015 SSR) devoted to the magnetotail current sheet as observed by Cluster. These publications provide statistical estimates for current sheet thickness, $L$, and current density intensity, $j_y$, for different geomagnetic activity level and different locations. I suggest Authors to use these (already published) materials to verify their model and address following questions: A) can modified Kan-like current sheet model (with $B_z \sim 1/x$) describe observed intensities of currents ($j_y \sim 5 - 10$ nA/m$^2$) for reasonable lobe magnetic field magnitudes and magnetotail configurations? B) How does current sheet thickness vary with $x$ (for different geomagnetic activity) and can model describe observed current sheet thicknesses?

(2) Authors demonstrated the model dependence on the rotational angle $\varphi$... but I believe readers would be interested to get more details about this dependence. How do distribution of current density and $B_z$ (in model with $B_z \sim 1/x$) depend on $\varphi$ in 2D $(x, z)$ plane? Can a finite $\varphi$ result in local (in $x$) current density increase? What are expected locations (along $x$) of current density intensifications for different $\varphi$ and how do these locations relate to Cluster (Petrukovich et al. 2009 JGR) and Geotail (e.g., Genestreti et al. 2014 JASTP) observations of the reconnection onset?

List of small changes:
(1) Abstract, line 7: I did not find any comparison with "realistic current sheets" in the

draft.

(2) page 1, lines 14-15: be more accurate here: Burkhart et al. 1972 should be 1992; Kuznetsova et al. 1995 did not provide numerical CS model, but use the analytical model for numerical simulations; Sitnov and Merkin 2016 describe analytical model. . . as well as Vinogradov et al. 2016;

(3) page 6, line 15-19: $B_z \sim 1/x^3$ is a property of chosen general class (3,4) of solutions of Eq. (1). Alternative solutions of Eq. (1) give $B_z \sim 1/x^\alpha$ with different $\alpha$ (see Vasko et al. 2013 PoP)

(4) I did not get an idea of the paragraph (page 7, line 30 – page 8, line 5). Please, rewrite it with more details.

---

## Referee Comment (RC2) · Anonymous Referee #2 · 13 Feb 2018

The paper extends an earlier study of Semenov et al., 2015 to justify the applicability of the earlier obtained solution of a bent current sheet in magnetotail studies. The paper is written clearly (except some minor problems listed below) and the paper subject is within the scope of the AG journal. Hence, I recommend to publish this paper after some revisions.

My only major comment is for the abstract. The first sentence of the abstract is confusing. As the authors say later on Page 2, in Lines 28-30, they actually don't develop any new solution. Rather they investigate applicability of an earlier-obtained solution to the magnetotail current sheet. The same is relevant for the first bullet in the "summary" on

[Figure]

Page 8.

Minor comments:

The term "asymmetric" when applied for a current sheet usually means that the plasma and magnetic field conditions are different at two sides of the current sheet, e.g., the magnetopause current sheet. The authors may want to use "curved current sheet" instead, or simply avoid "asymmetric".

Throughout the paper, please, choose the same sequence for Harris, Kan, Fadeev and Manankova when describing the family of solutions.

Page 1, Line 16: What is meant by "approximate" here?

Page 2, Lines 15-16: Please, split in two sentences.

Page 2, Lines 28-30: This should be in the abstract.

Page 3, Line 2: What is "current velocity"?

Page 3, Line 6: "typical length" of what?

Page 3, Lines 11-12, ", in general cannot be fulfilled" -> ", which is generally not fulfilled" or ", but is generally not fulfilled"? What is meant here?

Page 3, Line 24: What is "quasi-dipole"?

Page 4, Line 13: The tilt is $\varphi$ or $\varphi/2$?

Page 4, Line 15: Where should one look at Figure 2 to understand the contribution of "a"?

Page 5, Line 5: Would "isothermal" be observed in nature too?

Page 5, Line 8: "stops at" -> "stop near" (overshoot may happen).

Page 5, Lines 15-16: why these (x,z) are chosen for S calculations?

Page 7, Line 14: Faddev->Fadeev.

Page 8, Line 16: Unclear to what "yielding" refers to.

Figure 1: Please add axis units. "PHI" in the figure and $\varphi$ in the caption are different by a factor of 2 (also $\varphi$ in the text on Page 4, Line 13). Which is the correct one?

Figure 2: Please add axis units.

Figure 7: Figure caption is unclear. Color coding is unclear. Please, add axis units.

Figure 8: Please add axis units.

---

## Author Comment (AC1) · 2 Mar 2018

The authors thank reviewers for their help in improving the manuscript.

To address the reviewer's questions, the manuscript has been substantially complemented. Figures 8 – 11 are added (while Fig. 2 of the previous version is removed) and discussed in Page 7, 18 – Page 8, 19. Fig. 1 is redrawn and a number of minor corrections are also made. All corrections are marked red. Below, we address the referee's questions one-by-one.

**Response to Referee # 1**

**Comment:** can modified Kan-like current sheet model (with Bz ~ 1/x) describe observed intensities of currents (Jy ~ 5 – 10 nA/m$^2$) for reasonable lobe magnetic field magnitudes and magnetotail configurations?

**Response:** The comparison of the model scales (L, Jy) with the Cluster data reported in [Runov et al.(2006), Ann. Geophys., 24, 247] reveal the good match (with factor ~ 0.5 – 2) for current sheets with not too small values of ion number density, temperature  and velocity. The best match is detected for current sheets with typical scale L ~ 2000 – 8000 km and current density ~ 3 – 16 nA/m$^2$. See Figs. 10 and 11 in manuscript, and discussion in Page 8, line 3 – Page 8, line 19.

Notable, our Kan-like model does not reflect the CS bifurcation or embedding. In terms of the paper of [Petrukovich et al.(2015), Space Sci. Rev., 188, 311] this means that in our case $B_{ext} = B_0$. Hence, the model can usually reproduce the amplitude of current density, but not the profile Jy(z). This feature is illustrated in Figs. 2 and 3 of Runov et al. (2006). Harris profiles, shown there by dashed black curves are very similar to profiles obtained in our model. Thus, inaccuracy in current sheet width estimate is measured by factor 1 – 2, as well as current density.

Besides, the peaking current density is not equal exactly to normalization constant Jo, depending also on the model parameters and x-coordinate. To illustrate this dependence, we demonstrate profiles Jy(z) at x = 19 for several sets of parameters of the model (11 - 16) on the figure below.

[Figure]

**Figure 1**

Here, parameters a1 = a2 = f = 0, other parameters are given in the figure legend. The set of blue curves shows the dependence on parameter "b0", red set – the dependence on "n", and green set – the dependence on "phi". Dependence on parameter "k" is very weak, therefore it is not shown. It is seen that most efficiently profile Jy(z) is controlled by parameter "n". Though, one should keep in mind that the values of n > 1 produce configurations with the X-line, approaching the origin rather fast with increase of n.

Dependence on x is illustrated on Figure 4 below.

**Comment:** How does current sheet thickness vary with x (for different geomagnetic activity) and can model describe observed current sheet thicknesses?

**Response:** The second question is addressed above. Two figures below show profiles of Jy(z) at x = 19 for three levels of geomagnetic activity (see Fig. 4 of manuscript) for model parameter phi = 0, and phi = 60, respectively. Blue curves – quiet sheet, red curves – substorm, black curves – storm conditions.

[Figure]

Figure 2

[Figure]

Figure 3

It is seen that increase of the activity level reduces the sheet width, producing up to 20% thinning (estimating half-width of profiles at Jy = 0.5 Jy$_{min}$, we get ~ 0.9 for quiet conditions and ~ 0.7 for storm).

Sheet bending does not affect CS width, introducing vertical shift and small reduction of the current density.

The corresponding paragraph is added in Page 7, 24 – 28.

Next figure shows profiles Jy(z) for plane sheet, substorm conditions, in different cross-sections on x.

[Figure]

It is seen that current sheet is weakly thickening tailward: from 0.75 at x = 10, to 0.85 at x = 80.

**Comment:** How do distribution of current density and Bz (in model with Bz $\sim 1/x$) depend on phi in 2D (x; z) plane?

**Response:** Two figures (8 and 9) are added to manuscript to illustrate this dependence. Figures are discussed in Page 7, lines 18 – Page 8, 2. In a few words, tilt angle "phi" affects CS geometry, but does not affect its width. Parameter "n" affects sheet width and current density amplitude, but its variation is restricted from above, because X-point approaches the origin rather fast with increasing "n".

**Comment:** Can a finite phi result in local (in x) current density increase? What are expected locations (along x) of current density intensifications for different phi and how do these locations relate to Cluster (Petrukovich et al. 2009 JGR) and Geotail (e.g., Genestreti et al. 2014 JASTP) observations of the reconnection onset?

**Response:** Solutions without magnetic islands show monotonic decrease of $|Jy|$ in tailward direction before the X-line, and monotonic increase of $|Jy|$ behind the X-line, if any. Local peaks of current density are reproduced only in Fadeev-like models with an infinite chain of magnetic islands (non-zero parameter <f> in Eq.10). For example, figure below shows the quantity $Jy(x,z)$ for parameters {$a_1 = 0$, $a_2 = -0.03$, $b_0 = 22.13$, $k = 1$, $n = 1$, phi = 0} (plane sheet, substorm conditions) and $f = 0.003$.

[Figure]

**Figure 5**

And this figure shows profiles of Jy(x) at the sheet center

[Figure]

Figure 6

Reduction of the parameter "n" increases the spatial period of Jy variation (from 6.1 for n = 1.005 to 6.4 for n = 0.995 for current parameters).

Solutions with non-zero "f" may be used rather for some local analysis (e.g., Korovinskiy et al.(2018), PoP, 25, 022904), therefore we do not consider such solutions in the current paper, focused mainly on global characteristics.

**Minor comments and responses**

Abstract, line 7: I did not find any comparison with "realistic current sheets" in the draft.

Replaced by the term "averaged magnetic configurations".

Page 1, lines 14-15: be more accurate here: Burkhart et al. 1972 should be 1992; Kuznetsova et al. 1995 did not provide numerical CS model, but use the analytical model for numerical simulations; Sitnov and Merkin 2016 describe analytical model... as well as Vinogradov et al. 2016

Corrected, thank you.

Page 6, line 15-19: Bz ~ $1/x^3$ is a property of chosen general class (3,4) of solutions of Eq. (1). Alternative solutions of Eq. (1) give Bz ~ $1/x^\alpha$ with different $\alpha$ (see Vasko et al. 2013 PoP)

Thank you, the reference is added in Page 6, lines 23-24

I did not get an idea of the paragraph (page 7, line 30 – page 8, line 5). Please, rewrite it with more details.

Done. See Page 9, lines 13 – 23.

[revised manuscript text omitted]

---

## Author Comment (AC2) · 2 Mar 2018

The authors thank reviewers for their help in improving the manuscript.

To address the reviewer's questions, the manuscript has been substantially complemented. Figures 8 – 11 are added (while Fig. 2 of the previous version is removed) and discussed in Page 7, 18 – Page 8, 19. Fig. 1 is redrawn and a number of minor corrections are also made. All corrections are marked red. Below, we address the referee's questions one-by-one.

**Response to Referee # 2**

**Comment:** My only major comment is for the abstract. The first sentence of the abstract is confusing. As the authors say later on Page 2, in Lines 28-30, they actually don't develop any new solution. Rather they investigate applicability of an earlier-obtained solution to the magnetotail current sheet. The same is relevant for the first bullet in the "summary" on Page 8.

**Response:** Formally, in the current paper we develop a new solution, introducing the new parameter "n" in the generating function (10), affecting the geometry of the previously derived solution for bent current sheets [Semenov et al., 2015] and Bz magnetic component, as it is shown in Sec. 4. However, we agree, that the word "develop" used in the abstract may seem too pretentious, hence $1^{st}$ sentence of abstract and $1^{st}$ bullet in Summary are reworded as follows:
   a) Abstract: A specific class of solutions of the Vlasov-Maxwell equations, developed by means of generalization of the well-known Harris-Fadeev-Kan-Manankova family of exact two-dimensional equilibria, is studied. The examined model reproduces…
   b) Summary, bullet 1: An exact 2D bent CS equilibrium, built by means of generalization of the Harris-Fadeev-Kan-Manankova family of symmetric solutions of the Vlasov-Maxwell equations, is considered. The examined model reproduces the effects, related to the Earth dipole tilt and CS bending. The further generalization releases degeneracy of the original model, yielded too fast decrease of the normal magnetic component.

**Comment:** The term "asymmetric" when applied for a current sheet usually means that the plasma and magnetic field conditions are different at two sides of the current sheet, e.g., the magnetopause current sheet. The authors may want to use "curved current sheet" instead, or simply avoid "asymmetric".

**Response:** Indeed, the term "asymmetric CS" may be confusing somehow, therefore we replaced it by the term "bent CS". At the same time, we kept the term "asymmetric model" for the generalized Kan-like analytical solution. The proper corrections are made in text:

a) Page 2, 25: asymmetric CS → Page 2, 27, bent CS
b) Page 2, 29: analytical asymmetric solution → Page 2, 31, analytical solution
c) Page 2, 31: asymmetric CS → Page 2, 33, bent CS
d) Section 2 title: Asymmetric solution → Analytical solution
e) Page 4, 3: asymmetric (bent) CS → Page 4, 6, bent CS
f) Page 4, 5: asymmetric CS → Page 4, 8, CS
g) Page 4, 9: Asymmetric configurations → Page 4, 12, Configurations
h) Page 4, 16: asymmetric configurations → Page 4, 19, bent sheets
i) Page 5, 3: asymmetric CS → Page 5, 8, bent CS
j) Page 7, 15: asymmetric 2D CS → Page 8, 26, 2D bent CS
k) Page 7, 16: asymmetric CS → Page 8, 27, bent CS
l) Page 8, 14: asymmetric 2D CS → Page 9, 30, 2D bent CS

**Comment:** Throughout the paper, please, choose the same sequence for Harris, Kan, Fadeev and Manankova when describing the family of solutions.

**Response:** Historically, the proper sequence is Harris-Fadeev-Kan-Manankova. The proper corrections are made in text
a) Page 2, 29  and  b) Page 8, 25.

**Minor Comments and Responses:**

Page 1, Line 16: What is meant by "approximate" here?

Here, we talk about the solution of Panov et al. (2012), which is a pattern of Schindler-Birn family of approximate equilibrium solutions [e.g., Schindler and Birn (1978), Phys. Rep., 47, 109 – 165]

Page 2, Lines 15-16: Please, split in two sentences.

Done. Page 2, lines 17 – 18.

Page 2, Lines 28-30: This should be in the abstract.

But it is already there. Compare Page 2, lines 30 – 32 and Page 1, line 6.

Page 3, Line 2: What is "current velocity"?

Under the quasi-neutrality condition, in two-component plasma $\mathbf{j} = ne(\mathbf{V}_i - \mathbf{V}_e)$, hence the current velocity $\mathbf{V}_c = \mathbf{V}_i - \mathbf{V}_e$. We use this term following [Kan (1973), JGR, 78, 3773], who's model is considered here.

Page 3, Line 6: "typical length" of what?

Sorry, this is a typo. L is a typical scale of current sheet in the normal direction, the proper correction is made. Page 3, 9.

Page 3, Lines 11-12, ", in general cannot be fulfilled" -> ", which is generally not fulfilled" or ", but is generally not fulfilled"? What is meant here?

In general case of non-Maxwellian distribution functions condition (2) cannot be fulfilled. The explanation is corrected. Page 3, 15.

Page 3, Line 24: What is "quasi-dipole"?

We use the term "quasi-dipole" because this dipole-like solution possesses two extra singularities. See Fig. 1 and Page 4, lines 12 – 13.

Page 4, Line 13: The tilt is phi or phi/2?

The tilt is phi/2. To avoid any confusion, we state it explicitly in Page 4, 16, and in the caption of Fig. 1.

Page 4, Line 15: Where should one look at Figure 2 to understand the contribution of "a"?

Parameter "a" controls only the shift in vertical direction, as it is stated in Page 4, line 6. Therefore, we decided to remove Fig. 2 of the 1st version as low-informative.

Page 5, Line 5: Would "isothermal" be observed in nature too?

To the author's knowledge, it wouldn't. However, temperature variations across the single-peaked current sheets are not strong, $T_p$ varies for 10-20 %, see Fig. 5 in [Runov et al.(2006), Ann. Geophys., 24, 247].

Page 5, Line 8: "stops at" -> "stop near" (overshoot may happen).

That's right, thank you. Corrected on Page 5, 13.

Page 5, Lines 15-16: why these (x,z) are chosen for S calculations?

> As far as the analytical solution is compared to the T96 model, this particular interval in x direction is chosen due to the best in-situ data coverage (see Fig. 5 in Tsyganenko (1995), JGR, 100, 5599). The initial point of integration (30, -2.4) corresponds to the field line node.

Page 7, Line 14: Faddev->Fadeev.

> Thank you.

Page 8, Line 16: Unclear to what "yielding" refers to.

> It refers to "degeneracy of the original model". Replaced by "which yielded". Page 10, 1 – 2.

Figure 1: Please add axis units. "PHI" in the figure and phi in the caption are different by a factor of 2 (also phi in the text on Page 4, Line 13). Which is the correct one?

> Done. PHI in figure (tilt angle in clockwise direction) is equal to –phi/2 in analytical model.

Figure 2: Please add axis units.

> Figure is removed.

Figure 7: Figure caption is unclear. Color coding is unclear. Please, add axis units.

> Done.

Figure 8: Please add axis units.

> Done.

[revised manuscript text omitted]

---

## Author Response (AR1)

Dear editor,

We are, of cause, very pleased by your decision. On behalf of all co-authors I thank you for accepting the manuscript for publication.

Below, your questions/recommendations are addressed one-by-one.

1. **p3, line 14--> "...while in the general case..."**
   Corrected, thank you.

2. **p4, line 5 ---> "...for a bent..."**
   Corrected, thank you.

3. **Also, please clarify that in page 4, line 24, the x~340L is correct. Values of x quoted elsewhere in the manuscript (e.g Figures 1-9) are typically much smaller than 100L.**

   The value 340 L is correct. Here, we talk about the location of the X-point, which stays very far downtail (see Fig.2) for all reasonable (for the Earth) dipole tilt angles and parameters of the model (7-9). To avoid any confusion, we reworded this paragraph, see p.4, lines 22 – 27.

4. **Finally, while an expression of L to which results are normalised is given in eq. 17 (page 8), a similar one (but maybe with slightly different absolute value) is given in Line 8 of page 9. Can you verify that the normalisation of L is the same in both cases and if normalised x values quoted before and after equation 17 can be compared?**

   Yes, normalization expression, given in line 8 of page 3 (not 9) and in Eq. (17) are the same. To state it unequivocally, we include the definition, given on p.3, in Eq. (17). See p.8, line 5.